# Estimating the basic reproduction number at the beginning of an outbreak

**Sawitree Boonpatcharanon**[1], **Jane M. Heffernan**[2,3] *, **Hanna Jankowski**[2,3]

**1** Department of Statistics, Chulalongkorn Business School, Chulalongkorn University, Bangkok, Thailand, **2** Mathematics & Statistics, York University, Toronto, Canada, **3** Centre for Disease Modelling, York University, Toronto, Canada

☉ These authors contributed equally to this work.
* jmheffer@yorku.ca

**Data Availability Statement:** Yes - all data are fully available without restriction; The code and data can be found at https://github.com/hannajankowski/R0_estimators_data.

## Abstract

We compare several popular methods of estimating the basic reproduction number, $R_0$, focusing on the early stages of an epidemic, and assuming weekly reports of new infecteds. We study the situation when data is generated by one of three standard epidemiological compartmental models: SIR, SEIR, and SEAIR; and examine the sensitivity of the estimators to the model structure. As some methods are developed assuming specific epidemiological models, our work adds a study of their performance in both a well-specified (data generating model and method model are the same) and miss-specified (data generating model and method model differ) settings. We also study $R_0$ estimation using Canadian COVID-19 case report data. In this study we focus on examples of influenza and COVID-19, though the general approach is easily extendable to other scenarios. Our simulation study reveals that some estimation methods tend to work better than others, however, no singular best method was clearly detected. In the discussion, we provide recommendations for practitioners based on our results.

## Introduction

The basic reproduction number, $R_0$, (also called the basic reproductive ratio) is defined as the expected number of new infections produced by a single (typical/average) infectious individual, when introduced into a totally susceptible population. $R_0$ is used in epidemiological studies of infectious diseases to gauge how contagious/transmissible an infectious disease is: if $R_0 < 1$, the disease will die out, and if $R_0 > 1$ infection can increase in the population. It is also used to determine how effective vaccination or other disease mitigation strategies need to be in order to protect populations from infection.

At the outset of an infectious disease outbreak, an immediate goal is to determine $R_0$, so that public health and healthcare decision makers can be informed. For example, at the debut of the COVID-19 pandemic, reports of $R_0$ estimates were plentiful (see e.g. [1–6]). In the recent MERS-COV, 2009 H1N1, and 2003 SARS epidemics, there were also numerous studies of $R_0$ globally (see [7–19] for a small snapshot).

**Funding:** We note that our work was supported by the Natural Science and Engineering Research Council of Canada. The funders had no role in the study design, data collection, analysis, decision to publish, or preparation of the manuscript. The authors received no salary from funders.

**Competing interests:** The authors have declared that no competing interests exist.

There are many statistical and mathematical methods that can be used to estimate $R_0$ [20–29]. A main difficulty in $R_0$ estimation is that the methods often depend on data that is not available, or the methods suffer from collection and/or reporting, or other, bias. Different estimators utilize different approaches to deal with these difficulties. Broadly speaking, estimators can be classified as real-time (requiring little computation time) and non-real-time (requiring more extensive computation). Real-time estimators typically rely on simple epidemic models and/or simplifications of models in an attempt to remove dependence on unobservables (such as the Susceptible-Infectious-Recovered, a.k.a. SIR, compartmental modelling framework). Non-real time methods generally handle unobservables via Bayesian or Monte Carlo approaches, at the cost of computing time. Often, real-time methods also assume some prior knowledge of other parameters, such as the serial interval (SI). It is therefore important to study the effects of misspecification of the either the modelling framework or input parameters on these estimators. For example, suppose an $R_0$ estimator has been constructed to work within a SIR disease modelling framework. Infectious diseases, however, can include periods of infection that are not infectious. The infectious period can also be split into various stages of asymptomatic and symptomatic infection, which ultimately affect the case reporting rate to public health. Therefore, methods that are based on the SIR modelling framework can project erroneous estimates of $R_0$, and differences in $R_0$ estimates may simply reflect poor estimator structure or application to data that has been misspecified.

A recent study by [27] has discussed several nuances of different estimator methods that can affect $R_0$ estimates. The effect of misspecification is only touched on briefly. In this work, we compare six different estimators of $R_0$: four real-time estimators and two estimators which require longer computation times. The four real-time estimators are based on an SIR or similar framework, while the two other estimators can be tuned to extensions of the SIR model. We then simulate data generated from one of three compartmental epidemiological models, the SIR, SEIR, and SEAIR models that track susceptible (S), exposed (E), asymptomatically infectious (A), symptomatically infectious (I), and recovered (R) individuals in their modelling frameworks. We note that three of the real-time estimators assume that the serial interval is known, and therefore we also consider the situation when this serial interval is guessed incorrectly in these estimators. Our work thus studies the effect of compartmental model and/or serial interval misspecification on the real-time estimators. Moreover, non-real-time methods require specification of the epidemiological model by the investigator, and our work studies the effect of compartmental model misspecification on these.

The report of our findings is organized as follows. We first provide an introduction to three compartmental infectious disease models that we use to generate case data. Six $R_0$ estimators are then introduced, including a discussion of their underlying compartmental model structure assumptions. We then apply each estimator to data generated from the three compartmental models, and Canadian COVID-19 data for the provinces of British Columbia, Ontario, Quebec, and also for the country as a whole. Early epidemic dynamics are discussed using the inflection point (or turning point) in the epidemic growth curve, the point at which the curvature in the epidemic growth curve changes—early timepoints exist before this point. We employ parameter values representative of respiratory virus epidemics, and in particular, influenza and COVID-19 [30–35]. We note that while daily data may be sometimes available during an infectious disease outbreak, it may not be complete and can include a reporting delay. We thus have chosen to use weekly case reports. Weekly case report data is also typical to outbreaks of influenza, a respiratory virus, and a chosen pathogen of study.

## Methods

### Epidemiological models

We focus on three compartmental epidemiological models that form the basis of all infectious diseases models [36–38], the

**SIR**: **S**usceptible–**I**nfectious–**R**ecovered

**SEIR**: **S**usceptible–**E**xposed–**I**nfectious–**R**ecovered

**SEAIR**: **S**usceptible–**E**xposed–(**A**symptomatic Infectious)–(Symptomatic **I**nfectious)–
 **R**ecovered

models. The models are each composed of three to five compartments (with labels matching the model name). Individuals transition from one compartment to the next based on pre-specified random dynamics. Here, we assume that these distributions are exponential, and thus assume systems of ordinary differential equations (ODEs). We use the notation $\theta = (\beta, \sigma, \rho, \gamma)$ to denote the vector of parameters for the models, see Table 1 for details. The ODE systems for all three are provided in the S1 Appendix, as well as their corresponding flow diagrams. All models are considered without inclusion of demography, i.e. birth and death. The total population is fixed throughout the simulation and denoted by $N$ with initial values of $S_0$ and $I_0$ for $S$ and $I$ populations, respectively, and all others zero. Therefore, for all three models $N$ is equal to $S_0 + I_0$, and this is approximately equal to $S_0$ since $S_0 >> I_0$. For the SIR model, for all $t \geq 0$ it also holds that $S(t) + I(t) + R(t) = N$. Similarly, $S(t) + E(t) + I(t) + R(t) = N$ for the SEIR model and for the SEAIR model, $S(t) + E(t) + A(t) + I(t) + R(t) = N$.

Data is generated using the SIR, SEIR, and SEAIR compartmental model structures using a stochastic agent-based modelling framework implemented in C++. The simulations progress at the level of individual hosts in the applicable model disease status compartments. The simulation moves forward using "event times" that are assigned to each infected individual in the population and are determined by the compartment characteristics of which an individual is currently a member. Such event times correspond to infection events, when an infected individual transmits the infection to a susceptible, and times at which infected individuals progress

**Table 1. SIR, SEIR, SEAIR model parameters and values, $R_0$, serial interval.**

(a) Model contact rate notation

| model | parameter | | | |
|---|---|---|---|---|
| | $\beta$ | $\sigma$ | $\rho$ | $\gamma$ |
| SIR | $S \to I$: $\beta I(t)/N$ | | | $I \to R$: $\gamma$ |
| SEIR | $S \to E$: $\beta I(t)/N$ | $E \to I$: $\sigma$ | | $I \to R$: $\gamma$ |
| SEAIR | $S \to E$: $\beta I(t)/N$ | $E \to A$: $\sigma$ | $A \to I$: $\rho$ | $I \to R$: $\gamma$ |

(b) Model parameters, $R_0$, and serial interval

| model | $\theta$ | $R_0 = R_0(\theta)$ | serial interval |
|---|---|---|---|
| SIR | $(\beta, \gamma)$ | $\beta/\gamma$ | $1/\gamma$ |
| SEIR | $(\beta, \gamma, \sigma)$ | $\beta/\gamma$ | $1/\gamma + 1/\sigma$ |
| SEAIR | $(\beta, \gamma, \sigma, \rho)$ | $\beta/\gamma + \beta/\rho$ | $1/\gamma + 1/\sigma$ |

(c) Parameter values for simulations

| model | influenza 1 | influenza 2 | COVID-19 |
|---|---|---|---|
| SIR | (1/3, 1/5) | (1/3, 1/5) | (1/2, 5/26) |
| SEIR | (1/3, 1/3, 1/5) | (5/9, 1/2, 1/3) | (13/11, 1/3, 5/11) |
| SEAIR | (1/3, 1/3, 1/2, 1/5) | (5/12, 1/2, 1, 1/3) | (26/57, 1/3, 2/7, 5/11) |

to the next stage of infection or recover. The C++ model is based on previous work [39, 40]. Again, we note that all event times are assumed to be exponentially distributed with mean $1/\xi$ where $\xi$ refers to the model parameter associated with the same transition in the system of ordinary differential equations. See Table 1.

1000 agent-based model simulations are conducted for each of the SIR, SEIR, and SEAIR frameworks with parameters as given in Table 1. Model parameters were taken from the literature, and are representative of pandemic influenza ($R_0 \in [1.2, 7]$, serial interval $\in [1.5, 9.5]$) and COVID-19 ($R_0 \in [1.6, 3.4]$, serial interval $\in [4.2, 7.5]$) [3, 30–32].

- The first influenza (influenza 1 in Table 1) example parameters are such that $R_0 = 5/3$ for SIR and SEIR and $R_0 = 7/3$ for SEAIR. For this example, the serial interval is 5 days for the SIR model and 8 days for the SEIR and SEAIR models.

- The second influenza (influenza 2) example parameters are such that $R_0 = 5/3$ and the serial interval is 5 days for each of the SIR, SEIR, and SEAIR models.

- The COVID-19 parameters are such that $R_0 = 2.6$ and the serial interval is 5.2 days, again, for all models. The incubation period in the SEAIR COVID-19 model has a mean of 6.5 days [32].

  For each epidemic, the population size $N$ is set to 10, 001 where $S(0) = 10, 000$ and $I(0) = 1$.

## $R_0$ and the serial distribution

The serial distribution is the distribution from the time that an infected individual (the infector) becomes symptomatic, to the time when a person infected by the infector, the infectee, becomes symptomatic. For the SIR model, this is the same as the time spent in the *I* compartment, and in particular, the serial distribution is exponential with mean $1/\gamma$ when exponential distributions are assumed throughout the model [41]. We summarize the serial intervals for our models in Table 1 [41]. In the literature, the serial distribution may also be referred to as the serial interval, although this most often refers to the mean of the serial distribution, or alternatively, a range indicating highly likely values from the serial distribution. Here, we will use the convention that the serial interval refers to the mean of the serial distribution. For diseases such as influenza, it may be reasonable to assume that the serial distribution is known apriori. For other situations, such as new emerging diseases, such assumptions are less valid.

## Methods for estimating $R_0$

Many methods exist to estimate $R_0$. We refer to [29] for a recent review. If the transition rates in the compartmental models are known, then $R_0$ can be easily calculated using the formulas listed in Table 1. However, full transition rates are generally not known in practice, and hence statistical estimation methods are required. The main difficulty in estimation is that complete data is unavailable for the full epidemiological model. Here, we consider six different methods of estimating $R_0$. For simplicity, we name the methods WP, seqB, ID, IDEA, plug-n-play, and fullBayes in this work. A summary of the methods and their key properties is given in Table 2 for reference.

The first four (WP, seqB, ID, and IDEA) are real-time methods based on simplifications of the full ODE epidemiological models. This simplification is necessitated by the fact that the full data is unobservable. In these methods, estimation of $R_0$ is coupled with either estimation or prior knowledge of the serial distribution.

The two latter methods (plug-n-play and fullBayes) do not simplify the full epidemic models, but handle the issue of unobservable data by Monte Carlo simulation (plug-n-play method)

**Table 2. Summary of estimation methods for $R_0$.**

| method | summary |
|---|---|
| WP | White & Pagano Method, due to [42]. Serial distribution can be assumed known or can be estimated using MLE; method developed under branching process model; simple method which yields real-time estimates (when serial interval is unknown the method takes longer to compute). |
| seqB | Sequential Bayes Method, due to [43]. Serial distribution assumed known (only the mean is used); method developed assuming SIR model and uses sequential Bayes methods; simple method which yields real-time estimates. |
| ID | Incidence Decay Method (see [44]). Serial distribution assumed known (only the mean is used); method developed assuming an SIR model structure and uses least squares estimation. It is a simple method which yields real-time estimates. |
| IDEA | The Incidence Decay and Exponential Adjustment Method is presented in [44]. Serial distribution assumed known (only the mean is used); method developed assuming SIR model and uses least squares estimation; simple method which yields real-time estimates. IDEA uses a slightly more complex model for fitting than ID. |
| plug-and-play | Plug-and-Play Method. See [45]. Serial distribution assumed unknown; method selects one of SIR/SEIR/SEAIR model; implementations available though not real-time (depending on input selection). Generally, this approach fits the complete model using maximum likelihood and relying on Monte Carlo to fill in missing observations. The R-package, called POMP, is quite technical and can be difficult to implement [45]. |
| fullBayes | Full Bayes Method. See [46]. Serial distribution assumed unknown; method selects one of SIR/SEIR/SEAIR model; not real-time. this approach fits the complete model using maximum likelihood and relying on Monte Carlo to fill in missing observations. Can be quite technical in implementation. |

or Bayesian priors with MCMC used to handle estimation due to model complexity (fullBayes method). As such, these methods are more computationally intensive. These two methods estimate the unknown transition rate parameter vector $\theta$ in the epidemic model. They do not require any prior knowledge, including prior knowledge of the serial distribution. Indeed, since the methods result in estimates of $\theta$, these can then in turn be used to derive an estimate of the serial distribution. Furthermore, the methods assume prior knowledge of the epidemic model, in the sense that the user can decide whether the SIR, SEIR, or the SEAIR model is more appropriate for the particular disease. In contrast, the WP, seqB, ID, and IDEA methods all rely on simplifications, and are not able to allow for such tailoring.

Although the plug-n-play and fullBayes methods are more computationally intensive and not considered "real-time", we note that modern day access to computational power is blurring this line of distinction. Our implementations of fullBayes and plug-n-play were done on a non-specialized desktop computer and without special consideration to computing time in the implementations. The time required to obtain the estimates was less than two minutes in both cases, and we do not consider this to be prohibitive. Furthermore, more careful programming could yield even faster estimates. A more detailed discussion is available in Sectio. Computational Time.

**WP: Maximum likelihood estimation of a branching model.** [42] developed a straightforward estimation method whereby either the serial distribution is known, or the serial distribution is estimated along with $R_0$. The method assumes that only the number of infectious individuals at discrete time points (e.g. daily or weekly) is observable and both approaches (serial known and unknown) use maximum likelihood. Recall that $I(t)$ denotes the number of infecteds (i.e. the individuals in compartment $I$) at time $t$. Using our notation, and assuming that the times $t_0 = 0$, $t_1$, $t_2$, . . ., $t_\kappa$ are integers which count, for example, the number of days or weeks since the beginning of the pandemic (time zero), [42] obtain the log-likelihood

$$\ell(R_0, p) \quad = -\sum_{i=1}^{\kappa} \mu(t_i) + \sum_{i=1}^{\kappa} I(t_i) \, \log \, \mu(t_i),$$

where $\mu(t) = R_0 \sum_{j=1}^{\min(\kappa,t)} I(t - t_j)p(t_j)$ and $p$ is a vector denoting the (discrete and finite) serial distribution on $t_1, \ldots, t_\kappa$. That is, if $Y$ is the random variable representing the serial distribution then $p(t_j) = P(t_j \leq Y < t_{j+1})/P(Y \leq t_\kappa)$. If $p$ is known (notably, this includes knowing the value of $t_\kappa$ which describes the support of $p$) then the maximum likelihood estimate of $R_0$ is straightforward to compute. In the SIR model with exponential transitions, $p(t_j)$ is a truncated geometric distribution. If $p$ is unknown, then [42] recommend discretizing a gamma distribution to simplify estimation. Other models (SEIR and SEAIR) do not have simple closed form expressions for $p(t_j)$ (see [41]). We found that for coarse data (e.g. weekly) the discretization and mean dominates the values of $p$ more so than the actual distribution chosen.

The WP method assumes an underlying branching process, which is neither of the SIR/SEIR/SEAIR models from which our data sets are generated. This model assumes, in particular, that throughout, the population size "available" to be infected remains constant, which does not hold for our simulated ODE models. As such, estimates should only really be considered early on in the epidemic. In our simulations presented below, we highlight the inflection point of each epidemic, and the WP method should only really be considered valid before this time.

The method has been implemented in [47], see also [48] for details on the R package called $R_0$. In our simulations, we found this implementation to have some numerical instability issues, which is most likely caused by the particular parameters of our simulated data sets. This instability was particularly profound when $p$ was assumed unknown, and most often the algorithm would not yield a solution. For this reason, we programmed our own implementation, for which we used a simple grid search. The built-in alternative optimization function in R uses the bisection method, and was very sensitive to the starting value (a small change in the starting value could change the $R_0$ estimate by orders of a thousand). In comparison, the grid search approach performed better, although it was still not ideal. The likelihood surface is very flat, which resulted in a non-unique MLE (we report only a default value). This property of the likelihood surface is most likely what also causes the issues we observed for our data in the implementation of the $R_0$ R package [48].

Furthermore, note that the log-likelihood assumes that the serial distribution is discrete, and that this discretization matches the observed data. That is, if data is observed weekly, the serial distribution is only known *on a weekly timescale*. This discretization can affect the serial distribution considerably, particularly if the timescale is quite coarse.

**seqB: Equential Bayes estimation using an SIR approximation.** [43] developed a Bayesian approach used to estimate $R_0$. As above, it is assumed that infectious counts are observed at periodic times such as days or weeks. The basic idea is to start with a mildly informative prior on $R_0$ and then update sequentially. The approach is based on the SIR model, and assumes that the mean of the serial distribution is known (under the SIR model, this is equivalent to knowing the parameter $\gamma$ which is the inverse of the mean of the serial distribution). [43] note that under the SIR model, and considering time interval $t_{j+1} - t_j$

$$
\begin{aligned}
I(t_{j+1}) \quad &= I(t_j) \exp\left[\gamma \int_{t_j}^{t_{j+1}} \left(R_0 \frac{S(s)}{N} - 1\right) ds\right] \\
&\approx I(t_j) \exp\left[(t_{j+1} - t_j)\gamma(R_t - 1)\right],
\end{aligned}
$$

where $R_t = R_0 S(t)/N \approx R_0$ at the beginning of an infection. Using this result, seqB assumes that the conditional distribution of $I(t_{j+1})$, conditional on $I(t_j)$, $R_0$, is Poisson with mean $\lambda = I(t_j)$ $\exp\{(t_{j+1} - t_j)\gamma(R_0 - 1)\}$. In the approach, $\gamma$ is known, and a prior is placed on $R_0$. With $N_0$ also assumed known, posterior estimates are found using a hierarchical or sequential Bayes approach. Note that the method cannot handle data sets where there are no new infections

observed in some time interval $t_{j+1} - t_j$ (as this results in a Poisson mean of zero). Therefore, the times at which infectious counts are observed must be sufficiently coarse so that all counts are non-zero (e.g. weeks instead of days). The method would also be inappropriate for situations where long intervals between cases are observed in the initial stages of the epidemic. This was observed, for example, in Canada for the first cases of COVID-19.

Although the above development is based on the SIR model, the resulting approximation behaves similarly to a branching process, much like the WP method. We therefore again consider this estimator valid only in the early stages, which for our simulations translates to times prior to the inflection points of the epidemic.

The posterior distribution of $R_0$ will have the same support as the prior, and placing a discretized prior on $R_0$ makes computations relatively straightforward, since the normalizing constant of the posterior is easy to implement. In the R implementation in [48], called $R_0$, the initial prior on $R_0$ is assumed to be uninformative. Their package focuses on the posterior mode, and much like their implementation of the WP method, uses a discretized version of the serial distribution (which could affect the input value of $\gamma$). We again chose to use our own implementation, and report the posterior mean which minimizes the Bayes' risk.

**ID and IDEA: Least square estimation using incidence decay approximations.** [44] introduced two simplified models describing the relationship between $R_0$ and other epidemic parameters in the SIR model. The first of these is the incidence decay (ID) model where

$$\tilde{I}(s) = R_0^s. \tag{1}$$

In the model, time $s$ is measured in units re-scaled based on the serial distribution. Recall that under the SIR model the serial distribution is exponential with mean $1/\gamma$. We then have the relationship in (1) that $\tilde{I}(s) = I(\gamma s)$. As (1) is only valid for a short (and unknown) period of time, [44] proposed a second alternative formulation, where a decay factor $d$ was introduced in order to reflect the often observed outbreak decline. In the incidence decay and exponential adjustment (IDEA) model, the relationship becomes instead

$$\tilde{I}(s) = \left( \frac{R_0}{(1+d)^s} \right)^s. \tag{2}$$

Under the ID model, we can solve (1) to obtain

$$R_0 = \tilde{I}(s)^{1/s}.$$

Of course, this relationship is not valid for real data across all values of $s$ as $\tilde{I}(s)$ is stochastic. To obtain an estimate of $R_0$ least squares is a natural option, and hence the ID estimator is the minimizer of

$$\sum_{j=1}^{k} \left( \log R_0 - \frac{1}{s_j} \log \tilde{I}(s_j) \right)^2,$$

which yields

$$\exp \left\{ \frac{1}{k} \sum_{j=1}^{k} \frac{1}{s_j} \log \tilde{I}(s_j) \right\}. \tag{3}$$

As noted above, the number of infectious people increases rapidly at the beginning of an outbreak, so a method based on (1) is expected to underestimate $R_0$. The IDEA model was

introduced to overcome this issue. As in the ID model, we solve (2)

$$R_0 = \tilde{I}(s)^{1/s}(1+d)^s,$$

and use least squares estimation to obtain its estimate. The IDEA estimator is defined then as the minimizer of

$$\sum_{j=1}^{k} \left( \log R_0 - \frac{1}{s_j}\log \tilde{I}(s_j) - s_j \log(1+d) \right)^2.$$

Unlike in the ID model, we also need to obtain a minimizer of $d$ to solve the optimization problem, and hence we require $k \geq 2$. Minimizing, we obtain

$$\exp\left( \frac{\left(\sum_{j=1}^{k} s_j^2\right)\left(\sum_{j=1}^{k} \frac{1}{s_j}\log \tilde{I}(s_j)\right) - \left(\sum_{j=1}^{k} s_j\right)\left(\sum_{j=1}^{k} \log \tilde{I}(s_j)\right)}{k\sum_{j=1}^{k} s_j^2 - \left(\sum_{j=1}^{k} s_j\right)^2} \right). \tag{4}$$

Details of these calculations are given in the S1 Appendix. Note that the formula is not valid for $k = 1$.

Both the ID and IDEA methods are straightforward and estimate $R_0$ directly, as long as the mean of the serial distribution is known. The model was built under the SIR assumption. In our simulations we examine the effect of misspecification of the underlying epidemic model.

**plug-n-play: Maximum likelihood using sequential Monte Carlo for partially observed epidemics.** Maximum likelihood is one of the more popular approaches used to estimate unknown parameters in a statistical model. The general idea is to find the parameter set $\theta$ which maximizes the likelihood (probability model) evaluated at the observed data. The difficulty for our setting is that our compartmental models (see the discussion of the epidemiological models) rely on data which is unobservable. In particular, the models require that the exact times of infections are known while we observe only daily or weekly counts of infectious individuals. The WP method [42], which also uses maximum likelihood, gets around this issue by creating a simplified model with a likelihood which relies only on observable data. Another alternative, discussed in [49], is to maximize the full likelihood and fill in the unobservables using many Monte Carlo simulations in a way which matches the fixed observable data points. Such an approach is often referred to as "plug-n-play".

The plug-n-play inferential method of [49] is based on likelihood inference using sequential Monte Carlo of partially observed Markov processes (POMP), also known as hidden Markov models or state-space models. The plug-and-play terminology comes from the fact that inference is based on Monte Carlo simulations from the model and does not require explicit expressions of the transition probabilities, which can be quite complicated. The algorithm for this method has been implemented in the R package POMP [45]. This software package can be accessed from the comprehensive R archive network (CRAN), see also [50]. As mentioned previously, the basic idea is to generate complete epidemic data in a way which matches the observed weekly infectious observations. To simplify the implementation, complete continuous-time data is not generated but rather an approximation is generated with observations of all components at a discretized time-scale $\Delta t$ (single value selected by the user). These discretized epidemics are generated using sequential Monte Carlo methods. An estimate of $\theta$ is then obtained via maximum likelihood using iterated filtering. The implementation in [50] allows for the selection of the model SIR, SEIR, or SEAIR. We refer to [49, 50] for additional details. The algorithm returns estimates of $\theta$, as well as an estimate of $R_0$ derived via the

formula

$$R_0 = \beta \frac{\Delta t}{1 - e^{-\Delta t \ \gamma}},$$

regardless of the epidemiological model. We refer to the estimate thus obtained as the plug-n-play estimator. R code detailing our simulations and choices of input values is provided as S1 File.

**fullBayes: Bayesian inference for partially observed epidemics.** Similar to the plug-n-play approach of the previous section, this is a simulation approach in which the incomplete observed data is replaced with complete data via simulations. The main difference is that the complete data is generated by placing a prior on its distribution in a Bayesian inferential approach. Some examples of epidemiological inference under the Bayesian paradigm are described in [46].

In order to describe the method we need first to introduce some additional notation. We do this for the SEAIR model, as all other models are simplifications of this case. Recall that we have observed infection counts $I(t_1), \ldots, I(t_k)$ at times $t_1, \ldots, t_k$. Let $m$ denote the vector with $j$th element given by the cumulative sums $m_j = \sum_{i=1}^{j} I(t_i)$. As such, $m$ describes the entirety of the observed data. For a time interval $[0, T]$ the complete epidemic includes much more information. Let $\tau_i^E, i \geq 2$ denote the individual times of exposure. Similarly, $\tau_i^A, i \geq 2; \tau_i^I, i \geq 2, \tau_i^R, i \geq 1$ denote the individual times of transitions into the asymptomatic, infectious, and recovered states, respectively. We assume that $m_0 = 1$. We also assume that all people who are infected in week $j$ will recover in week $j + 1$. Furthermore, we assume that the number of exposed and asymptomatic people in week $j$ is also equal to $m_j - m_{j-1}$. We let

$$\tau = \{\tau_i^A, i \geq 2; \tau_i^I, i \geq 2; \tau_i^R, i \geq 1\}$$

denote the epidemic path which contains all of this information.

As in [46], the first infection $\tau_1^I$ is treated separately as a parameter of the model. Hence a prior $\pi_I(\tau_1^I)$ is placed on this variable. Recall that $\theta$ denotes the vector of compartmental model parameters; see Table 1, (b) An independent prior is also placed on $\theta$, $\pi(\theta)$, and samples from the posterior distribution $\pi(\theta, \tau_1^I, \tau | m) \propto L(\theta, \tau_1^I | \tau, m)\pi(\theta)\pi_I(\tau_1^I)$ are obtained. The marginal distribution of $\pi(\theta, \tau_1^I, \tau | m)$ is $\pi(\theta | m)$, which is the posterior distribution of $\theta$ given the observable data, and the distribution we are interested in.

We now calculate the likelihood $L(\theta, \tau_1^I | \tau, m)$ for the SEAIR model.

$$L(\tau, m | \theta, \tau_1^I)$$

$$= \left\{ \prod_{i=2}^{m_k} \frac{\beta S(\tau_i^E)}{N} \left( I(\tau_i^E) + A(\tau_i^E) \right) \right\} \left\{ \prod_{i=2}^{m_k} \sigma E(\tau_i^A) \right\} \left\{ \prod_{i=2}^{m_k} \rho A(\tau_i^I) \right\} \left\{ \prod_{i=1}^{m_{k-1}} \gamma I(\tau_i^R) \right\}$$

$$\times \ \exp\left\{ - \int_{\tau_1^I}^{t_k} [\beta S(t)(I(t) + A(t))/N + \sigma E(t) + \rho A(t) + \gamma I(t)] dt \right\}.$$

The joint prior distribution of the unknown rate parameters $\theta$ is made up of independent gamma distributions given by $\Gamma(\alpha, k)$ with mean $k/\alpha$. We assume that $\alpha$ is the same for the parameters $\beta, \sigma, \rho, \gamma$, while $k$ varies and if appropriate will be denoted by $k_\beta, k_\sigma, k_\rho, k_\gamma$. In the simulations we take $\alpha = 1$ and $k_\beta = k_\sigma = 3, k_\rho = 2, k_\gamma = 5$. The prior distribution on $-\tau_1^I$ is exponential with rate one, and this is independent from the $\theta$ vector. Calculations given in the S1 Appendix give the posterior marginal distributions for $\pi(\tau_1^I | \theta, \tau, m)$ and $\pi(\theta | \tau, m, \tau_1^I)$ all of which have gamma distribution with closed form expressions for the parameters. Some

sensitivity analysis to the prior distributions was conducted (see S1 Appendix), and changing the prior did not visibly affect the results.

The general approach we take is now described using the following steps.

1. Use Markov chain Monte Carlo (MCMC) to simulate from $\pi(\theta, \tau, \tau_1^I | m)$.

2. From Step 1, we obtain a sequence of samples $(\tau_l, \theta_l, \tau_{1,l}^I)$ for $l = 1, \ldots, b + B$ from the posterior distribution $\pi(\theta, \tau_1^I, \tau | m)$. Here, $b$ denotes the burn-in period for the MCMC results, and $B$ denotes the number of MCMC samples collected. To obtain an estimate of $\theta$, from the samples $l = b + 1, \ldots, b + B$, one option is to simply average the values $\theta_l$. Instead, we treat each $(\tau_l, \theta_l, \tau_{1,l}^I)$ a sample from the full posterior model, and calculate the posterior mean of $\bar{\theta}_l$, using the formulas given in the S1 Appendix.

3. Average the posterior means $\bar{\theta}_l, l = b + 1, \ldots, b + B$ to obtain an estimate of $\theta$.

The final reported estimate is obtained from the estimate of $\theta$ in Step 3 using the appropriate formula in Table 1. In our simulations, we take $b = 100$ and $B = 1000$, and refer to the estimator as fullBayes.

The MCMC algorithm we use is the Metropolis-within-Gibbs. Namely, there are three main components to the posterior distribution $\theta$, $\tau$, and $\tau_1^I$. In the S1 Appendix, the posterior distributions for $\pi(\theta | \tau, \tau_1^I, m)$ and $\pi(\tau_1^I | \theta, \tau) = \pi(\tau_1^I | \theta)$ are obtained in closed form. Given one observation of $(\theta_l, \tau_l, \tau_{1,l}^I)$, the algorithm generates the next observation as follows.

1. Sample $\tau_{1,l+1}^I$ from the posterior $\pi(\tau_1^I | \theta_l)$.

2. Sample $\theta_{l+1}$ from the posterior $\pi(\theta | \tau_l, \tau_{1,l+1}^I, m)$

3. Sample $\tau_{l+1}$ using a Metropolis step:

 (a). Propose a new $\tau$: For each $i = 1, \ldots, k$

 (i). $\tau_j^E$ is IID uniformly distributed on $[t_{i-1}, t_i]$ for $j = m_{i-1}, \ldots, m_i$

 (ii). $\tau_j^A$ is IID uniformly distributed on $[t_{i-1}, t_i]$ for $j = m_{i-1}, \ldots, m_i$

 (iii). $\tau_j^I$ is IID uniformly distributed on $[t_{i-1}, t_i]$ for $j = m_{i-1}, \ldots, m_i$

 (iv). $\tau_j^R$ is IID uniformly distributed on $[t_{i-1}, t_i]$ for $j = m_{i-2}, \ldots, m_{i-1}$

 (b). Accept the proposal with probability $\min\{1, \alpha\}$ where

$$\alpha = \frac{\pi(\tau | \theta_{l+1}, \tau_{1,l+1}^I, m) g(\tau | \tau_l)}{\pi(\tau_l | \theta_{l+1}, \tau_{1,l+1}^I, m) g(\tau_l | \tau)} = \frac{L(\tau | \theta_{l+1}, \tau_{1,l+1}^I, m)}{L(\tau_l | \theta_{l+1}, \tau_{1,l+1}^I, m)},$$

 noting that with the proposal distribution in (a), we have that $g(\tau | \tau_l)/g(\tau_l | \tau) = 1$. Details are provided in the S1 Appendix

The chain is initialized by sampling $\theta$ from its prior distribution.

## Real world COVID-19 data

We consider an example for the COVID-19 pandemic in Canada. The first case of COVID-19 was recorded on January 25th, 2020 in Toronto, Ontario [51]. For the first few weeks, isolated cases arrived, however strict contact tracing kept the pandemic from beginning. We therefore do not consider the first four weeks of the pandemic timeline (there were very few cases, and most weeks had zero cases at this stage). In late February, the pandemic took hold and cases

began to grow exponentially with community transmission [51]. Approximately one month from this, non-pharmaceutical measures were imposed and most provinces went into some form of lockdown. We therefore do not consider data much longer after lockdown initiation as these measures would decrease the transmission rate.

We estimate $R_0$ for all of Canada, and for the three most populous provinces, British Columbia (BC), Ontario, and Quebec. In Ontario, strict restrictions were imposed following March break (a one week school break during the winter) which fell around March 20th, 2022. In Quebec, lockdown was imposed around March 24th, and strict public measures were implemented around March 17th in BC. Epidemic data is provided from [52]. Public health mitigation data and dates are provided by [51].

## Workflow

The goal of our study is to quantify $R_0$ estimation in well-specified and misspecified settings, including misspecification of the model and serial distribution. For all models we therefore consider data coming from SIR, SEIR, and SEAIR epidemiological models, and the realworld COVID-19 pandemic in Canada. We study the $R_0$ estimation methods as follows:

Using synthetic data provided by the SIR, SEIR, and SEAIR models, we apply the following methods for well-specified and misspecified settings

1. WP method assuming

   - serial distribution (SD) is known and set to exponential with correct mean (5 days for influenza 1 and 2 and 5.2 days for COVID-19)

   - SD is known and set to exponential with incorrect mean (3 days for influenza 1, 2 and 7 days for influenza 2, and 4.2 and 7.5 days for COVID-19)

   - SD is unknown and estimated from a gamma distribution with unknown mean and variance (using a grid search algorithm)

2. seqB method assuming

   - SD has the correct mean (5 days for influenza 1 and 2 and 5.2 days for COVID-19)

   - SD has an incorrect mean (3 days for influenza 1, 2 and 7 days for influenza 2, and 4.2 and 7.5 days for COVID-19)

3. ID and IDEA methods assuming

   - SD has the correct mean (5 days for influenza 1 and 2 and 5.2 days for COVID-19)

   - SD has an incorrect mean (3 days for influenza 1, 2 and 7 days for influenza 2, and 4.2 and 7.5 days for COVID-19)

4. plug-n-play and fullBayes methods developed assuming

   - SIR

   - SEIR (SEIR and SEAIR data only)

   - SEAIR (SEAIR data only)

In these examples, the outbreaks are followed for 15 weeks, and this is the timeline given in our results. This timeline is presented only as a comparison to what is happening at the earliest stages. It also, however, improves the comparison between methods. Our comments below

focus only on the time period before the inflection point (denoted as a vertical blue line for all methods).

Using real world data, we apply the WP, seqB, ID, and IDEA methods with known SI, using incorrect and true values for COVID-19. We then apply WP, fullBayes and plug-n-play. Estimates are generated using weeks 5 to 10 for Canada, BC, Ontario, and Quebec. The date that lockdown was implemented is indicated by a vertical line for all three provinces. No such line is given for all of Canada, as the measures were handled provincially and not nationally.

When considering the results, recall that seqB and IDEA methods require at least two weeks of observations.

## Results

### Epidemic simulations

Fig 1 plots the number of individuals in compartment $I$ for each model structure, and each parameter set. The grey lines plot the simulation outcomes while the black lines plot the mean of the simulation data. Although the complete epidemic path is simulated, we assume that only the weekly number of infectious people is actually available. The epidemics are followed for 15 weeks, which covers the first 100 days of an outbreak. Simulation data is recorded at every event time. Weekly data is extracted from each simulation and saved in a data file for use for all of the $R_0$ estimators employed here. The blue vertical line indicates the point of inflection, where the concavity/curvature of the black line changes. The inflection points are 7, 12, and 9 for influenza 1 parameter values, 6, 7, and 7 weeks for influenza 2, and 3, 5, and 6 weeks for COVID-19, for the SIR, SEIR, and SEAIR models, respectively. These points are used to determine appropriate time intervals for $R_0$ estimation for each model since $R_0$ estimates are associated with early exponential growth and can be affected by decreases in the growth rate as the epidemic continues towards and past the point of inflection. Thus, "early in the epidemic" is the same as prior to the point of inflection. In real data, this time point would be unknown. Code and files containing all results have been provided in the S1 File.

### $R_0$ estimates

**Using synthetic data from the SIR, SEIR and SEAIR epidemiological models.** We summarize our numerical results in plots comparing the average mean squared error (MSE), side-by-side boxplots, as well as tables reporting the median $R_0$ estimates and its standard deviation. Again, these are all provided in a separate file as S1 File. In the main manuscript, we show only plots comparing the MSE of the various methods for the SIR data for the influenza 1 and 2 examples (Figs 2, 3 and 5), and SEAIR for the COVID-19 example (Figs 4 and 5). The MSE plots do not include the WP method where the serial distribution is estimated, as here the MSE was much too large to report. This can be ascertained from the Tables and the side-by-side boxplots provided in the Supplementary Material (in particular, see Tables 7, 12 and 17 in S1 File).

Figs 2 and 3 plot the MSE of the estimated $R_0$ values and the true $R_0$ value, for the WP, seqB, ID, and IDEA methods for the influenza 1 and 2 examples, using SIR data, and assuming a known serial interval. These plots provide examples of the well-specified and misspecified cases, using the true and misspecifed values of the known serial interval. Of the methods presented in these plots, seqB performs best, followed by ID. When SEIR and SEAIR data are considered, all estimators have larger MSE. However, our conclusion does not change (se. Sections 1–3 of the additional file included as S1 File) and seqB and ID still perform best. Finally, considering both bias and variance, as shown in the totality of boxplots and tables in the S1 File, our conclusion remains the same.

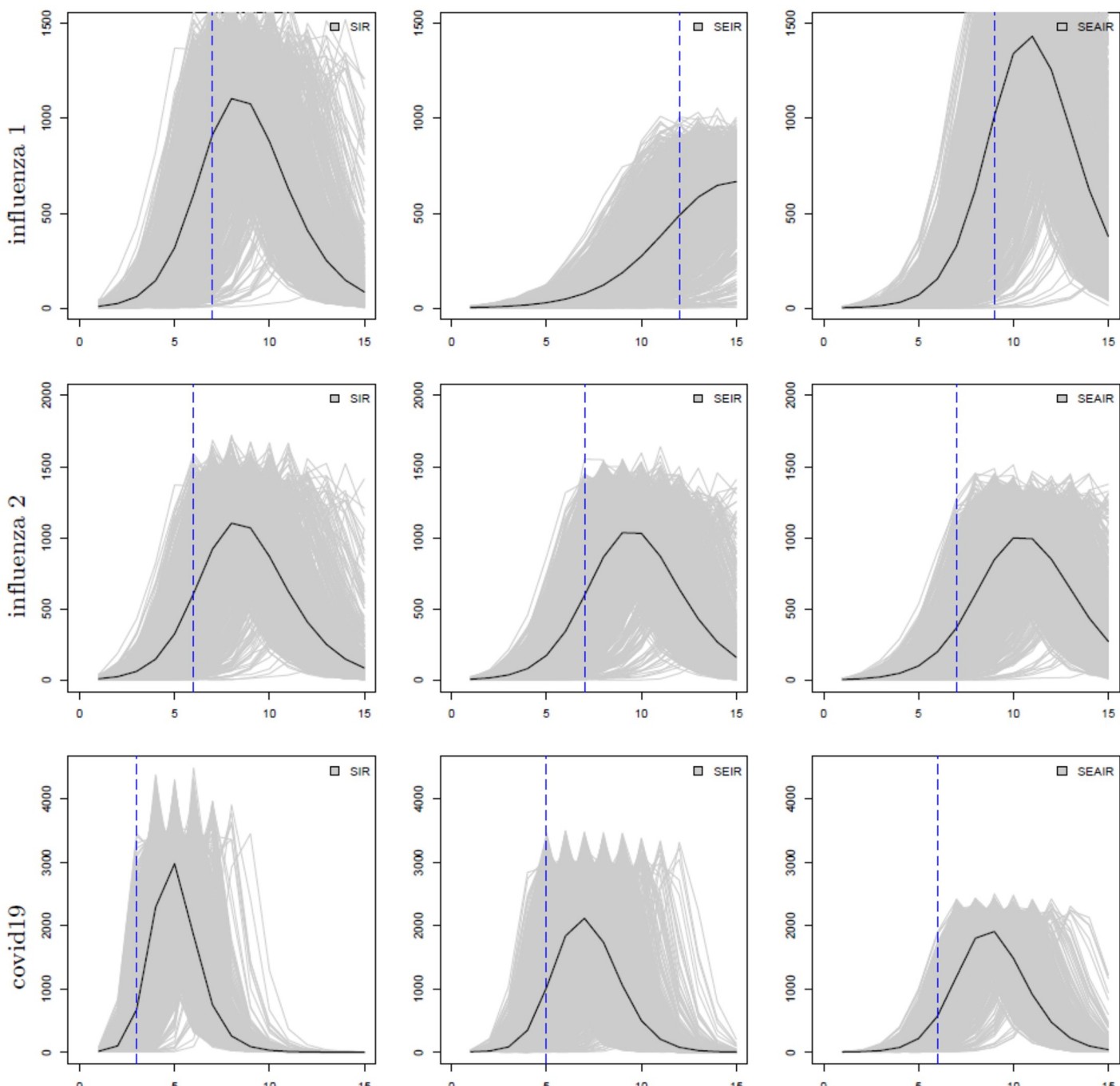

**Fig 1. The number of infectious individuals (*y*-axis) at time *t* in weeks (*x*-axis); from left to right: SIR, SEIR, and SEAIR; from top to bottom the examples are influenza 1, influenza 2, then covid19.** Individual simulated outbreaks from 1000 simulations are shown as grey lines, and their average is denoted as a black line. The blue vertical dashed lines show the inflection points for each model.

Fig 4 plots the MSE of the estimated $R_0$ values and the true $R_0$ value for the WP, seqB, ID and IDEA methods for the COVID-19 example, using SEAIR data. These plots provide examples of misspecification given incorrect serial interval (serial intervals of 4.2 and 7.5 days are incorrect, and 5.2 days is the true value), and given misspecified data where SEAIR

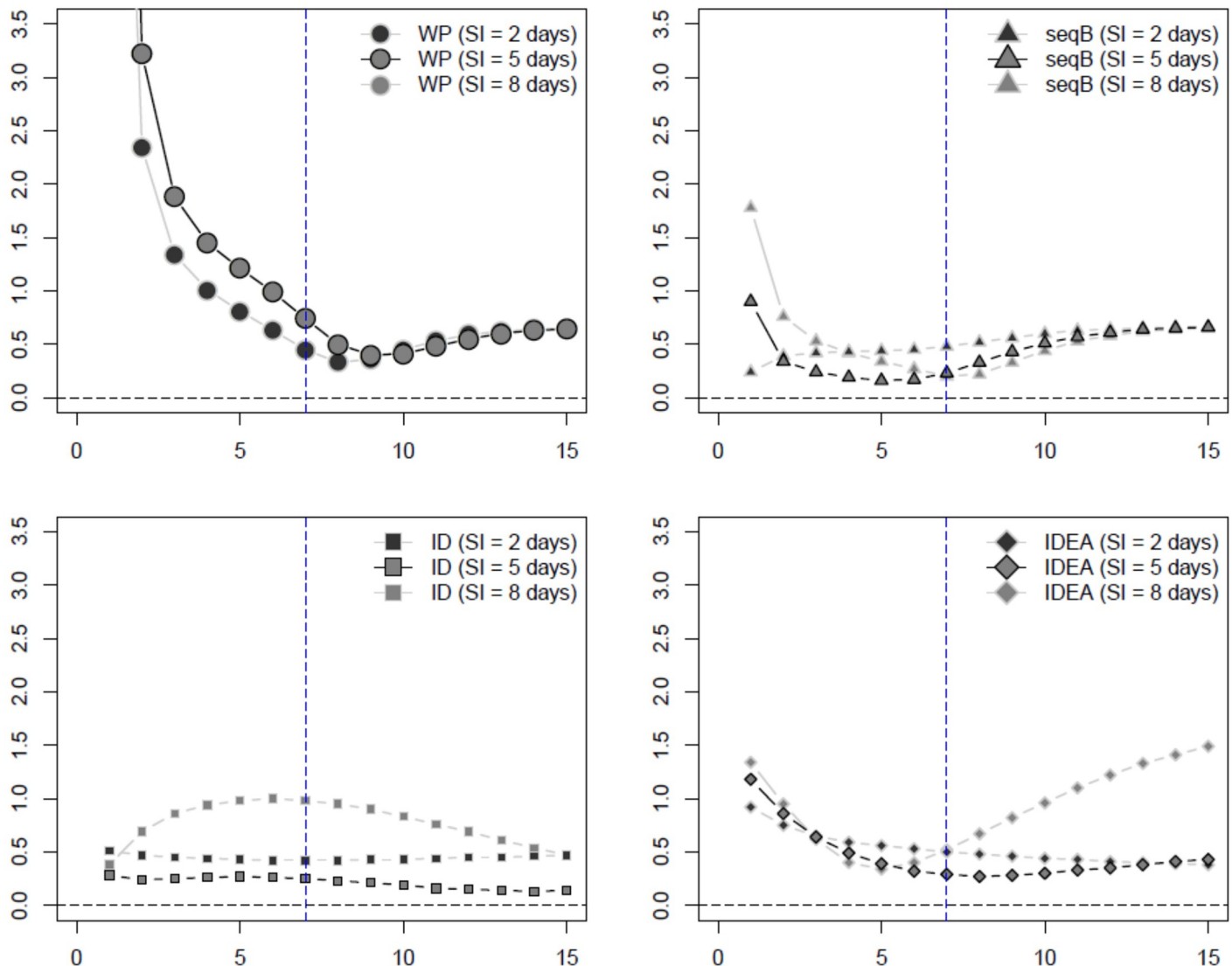

**Fig 2. Influenza example 1 estimated MSE of $R_0$ estimators assuming known serial interval (SI) with SIR data (week on $x$-axis).** The inflection point indicated by the blue dashed vertical line.

data is used for these methods that relate best to the SIR model framework. Here, again, seqB performs best, followed by ID. This is also true when SIR and SEIR data are considered, and considering bias and variance as presented in the totality of boxplots and tables in the S1 File.

We plot the MSE of $R_0$ estimates calculated using the fullBayes and plug-n-play methods in Fig 5 for influenza 1 and 2 examples using SIR data and SIR model structure, and for the COVID-19 example using SEAIR data, but with SIR, SEIR and SEAIR model structures. In all cases presented in this figure, we find that plug-n-play outperforms fullBayes. fullBayes performs well in the longterm, but this is not our goal—$R_0$ estimates are needed early on in the epidemic. A review of all of the cases presented in the S1 File confirm our conclusion.

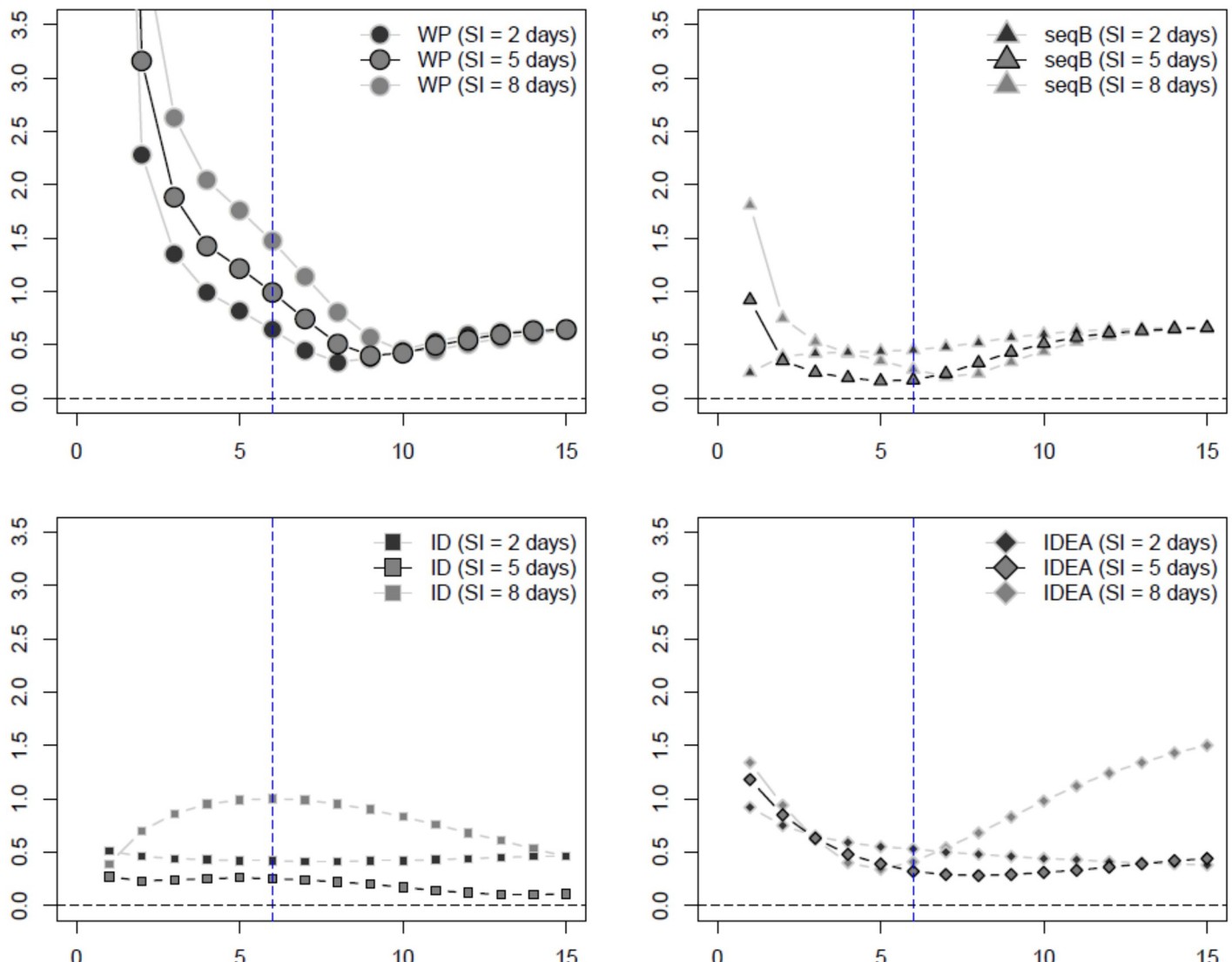

**Fig 3. Influenza example 2 estimated MSE of $R_0$ estimators assuming known serial interval (SI) with SIR data (week on $x$-axis).** The inflection point indicated by the blue dashed vertical line.

Computational time is a crucial factor as real-time estimates are desirable. Table 3 shows computational time for the SEIR model for a single data set and using a 1.60GHz/8GB RAM 64-bit operating system, x64-based processor. The results in this work are based on fullBayes with 1000 iterations and plug-n-play with 1000 particles and 10 IF iterations, where IF stands for the iterated filtering algorithm. The fullBayes method was implemented in R, and it is possible that faster implementations can be achieved using a different programming language. In comparison, the real-time methods (WP, seqB, ID, and IDEA) take less than one second each to compute.

Based on the estimator outcomes, our recommendations are as follows. When the serial interval is known, we recommend seqB and ID. We also recommend plug-n-play when the serial interval is known. When the serial interval is unknown, plug-n-play performs the best. Overall, we recommend that a suite of these estimators be used—employ plug-n-play, seqB,

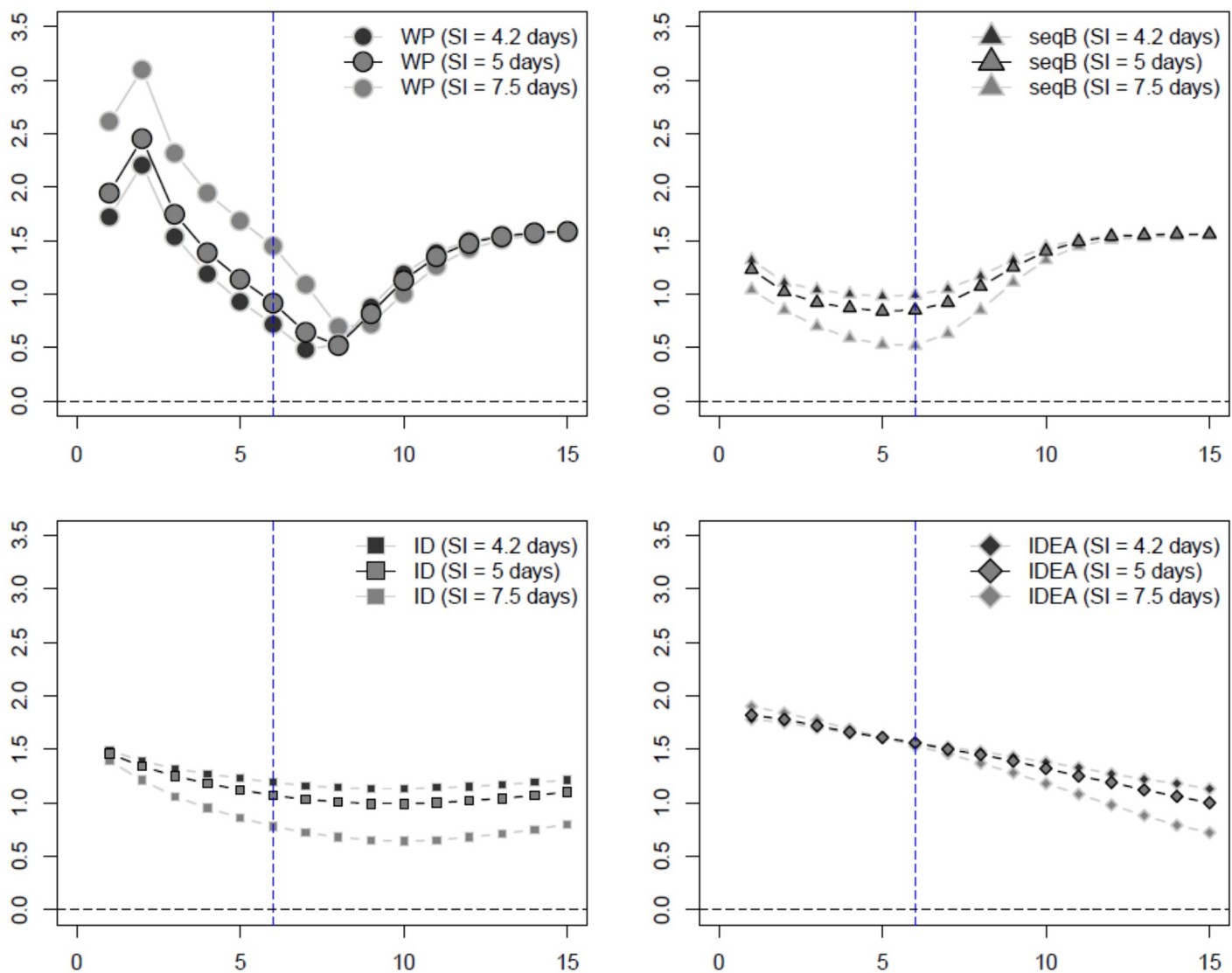

**Fig 4. COVID-19 estimated MSE of $R_0$ estimators assuming known serial interval (SI) with SEAIR data (week on *x*-axis).** The inflection point indicated by the blue dashed vertical line.

and ID. When the serial interval is unknown, a range of serial intervals can be provided to the seqB and ID methods to compare to the plug-n-play results. Practitioners, however, should consider their own preferences as to bias and variability of the estimators. We note here that as this study is focused on data observed weekly, our results may not be applicable to data observed, for example, daily, as the effect of the serial distribution on the results may be different. We also assumed that our data did not suffer from collection bias, under-reporting, and reporting delay. These issues are important, but beyond the scope of this work. However, it is our belief that weekly data, as considered here, is less sensitive to some of these issues than more fine-grained data.

**Using real world COVID-19 data.**    Fig 6 shows plots of estimates of $R_0$ for all six estimators as applied to real world COVID-19 epidemic data from Canada. The provinces of BC (second column), Ontario (third column), and Quebec (last column) are studied, as well as the

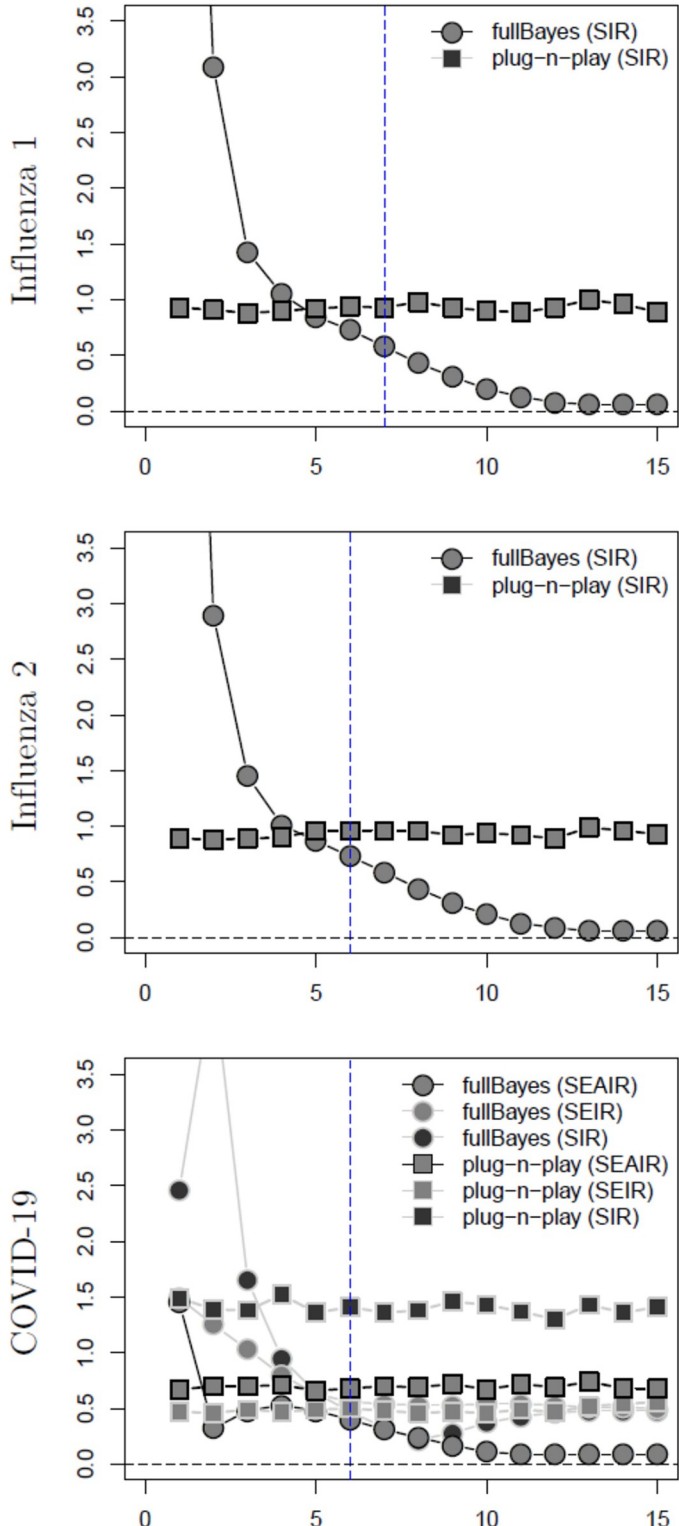

**Fig 5. Estimated MSE of $R_0$ estimators assuming unknown serial interval (SI) (week on $x$-axis).** For both influenza examples the data is SIR while for the COVID-19 example the data is SEAIR. The inflection point indicated by the blue dashed vertical line.

**Table 3. Computational time for the SEIR model for one data set (IF: Iterated filtering algorithm).**

| method | iterations | time | |
|---|---|---:|---|
| fullBayes | 1000 iterations | 8 | minutes |
| | 3000 iterations | 19.76 | minutes |
| plug-n-play (1000 particles) | 5 IF iterations | 3.10 | minutes |
| | 10 IF iterations | 5.82 | minutes |
| | 100 IF iterations | 58.44 | minutes |
| | 1000 IF iterations | 9.77 | hours |

entire nation (first column). The WP, seqB, ID and IDEA methods are applied using assumed known serial intervals of 2, 5, and 8 days. We compare our estimates to previously found $R_0$ estimates (black horizontal lines) of the Canadian pandemic in reference [3], to the Greater Toronto Area (which represents approximately 1/6 of the Canadian population). In summary, seqB, ID and plug-n-play estimates perform best. seqB produces estimates within the range denoted by the black horizontal lines for all serial interval values considered. The same is true for early estimates for plug-n-play. The ID method achieves the lower estimate for all geographic jurisdictions. It is sensitive to the choice of serial interval value, however, and higher serial interval values may drive the estimation to lie above the upper bound. See, for example, the subplots for Canada and Ontario. Given the findings here, we again recommend a combination of seqB, ID, and plug-n-play methods for estimation of $R_0$.

## Conclusion

The basic reproduction number, $R_0$, is an important parameter for estimation early in an epidemic so that public health interventions can be informed. As many estimators exist, and the assumptions of the estimators as well as their dependency on particular biological estimates (i.e., the serial interval), vary between methods, it is expected that $R_0$ estimates will differ. It is thus important to understand what estimators provide better outcomes under both true and misspecified conditions. Since respiratory viruses (especially influenza, and coronaviruses i.e., COVID-19 of late) affect the global population every year, we have chosen to study the estimators of $R_0$ for these types of infections, which are typically modelled using SIR, SEIR and SEAIR compartmental models. We have also chosen to consider weekly case data, as this is characteristic of pandemic influenza and other pandemic respiratory infection outbreak reported data, globally (with the exception of COVID-19, which was reported almost daily in most regions until early 2022).

We have considered six estimators that are commonly used when determining $R_0$ for any infectious disease outbreak. We discussed the advantages and disadvantages of each method, including dependencies on proper estimates of the serial distribution, and the computational resources needed to run each estimator. Our simulations consider a variety of well- and misspecified settings. Briefly, we find that the WP method can provide close estimates to the true $R_0$ value if the SD is known, but when the SD is unknown, the method suffers greatly (see Tables 7, 12 and 17 in S1 File). The seqB method performs well given SIR data but underperforms if there is any misspecification; the ID and IDEA methods, are useful due to their simplicity. ID outperforms the IDEA model, but ID estimates of slightly higher MSE copared to seqB. fullBayes estimates can have large variabilities, and are sensitive to the underlying model structure, but the plug-n-play method provides consistent estimates even with only one week of data.

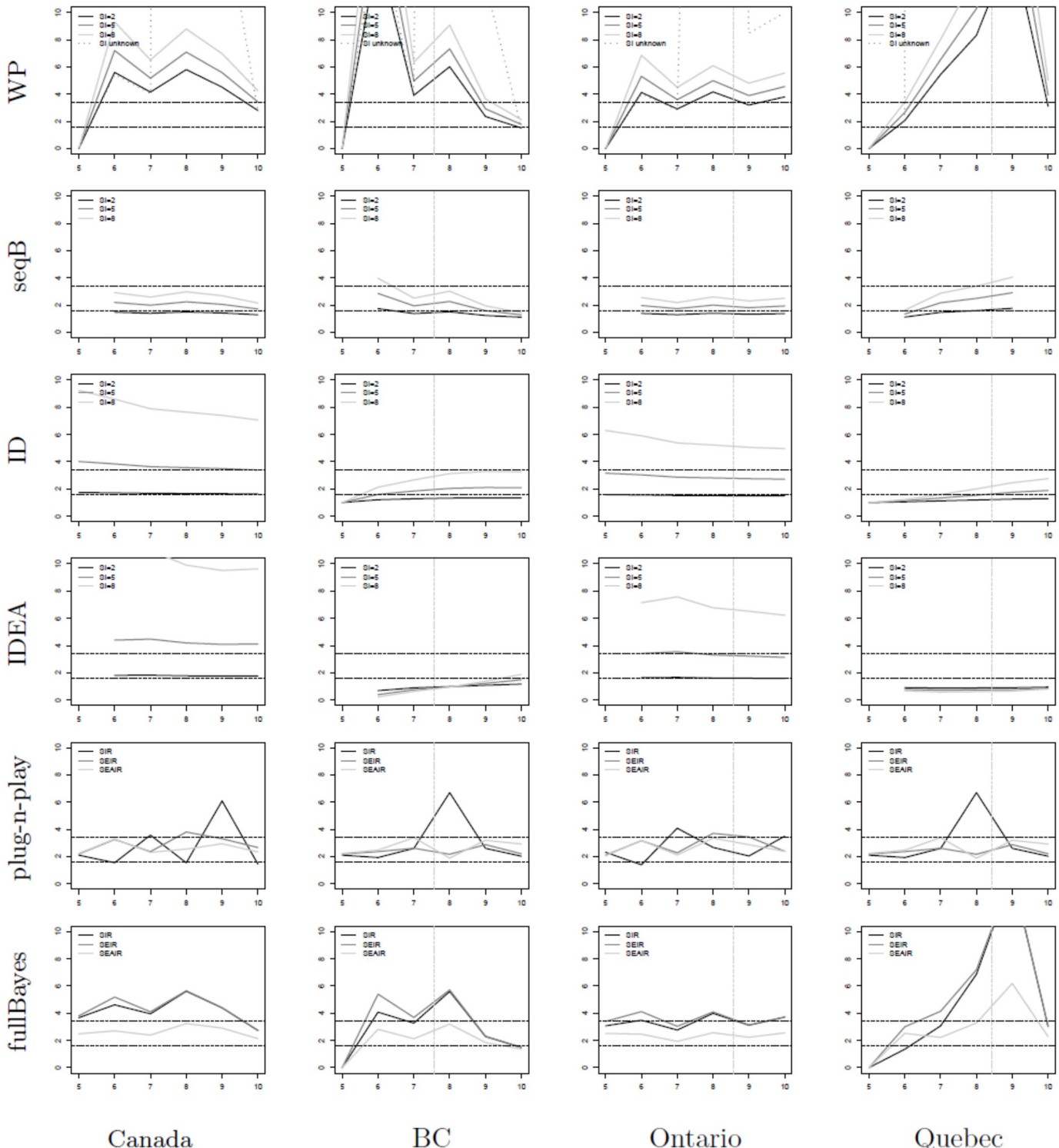

**Fig 6. $R_0$ estimators (*y*-axis) for COVID-19 data in Canada.** Data from [52]. The *x*-axis shows time in weeks where $t = 0$ denotes January 25, 2020—the date of the first known case in Canada [51]. The vertical gray line shows the date of lockdown for each of the provinces (there was no national lockdown date) [51]; while the horizontal lines denote estimates of $R_0$ from reference [3]. The provinces of BC (second column), Ontario (third column), and Quebec (last column) are studied, as well as the entire nation (first column). The WP, seqB, ID and IDEA methods are applied using assumed known serial intervals of 2, 5, and 8 days.

Considering both bias and variability, as well as misspecification, we find that the performance of the seqB, ID, and plug-n-play estimators is best, providing estimates of $R_0$ that are closest to the true value under both correctly specified and misspecified cases. Notably, plug-n-play does not require prior knowledge of the serial distributions. However, if the serial interval is known, seqB and ID outperform plug-n-play. Furthermore, seqB and ID require less computational time, and are easier to implement.

The choice of $R_0$ estimator is ultimately up to the practitioner. In our analysis we have shown that some $R_0$ estimators can be greatly affected by even a small level of misspecification. Given that biological certainty may be lacking at the beginning of an infectious disease outbreak, the number of disease stages needed in a model and a proper distribution of the serial interval may not be known. This means that a range of $R_0$ results will ensue, and the accuracy of the estimates will be unclear. We therefore recommend that a suite of estimators be used when estimating $R_0$. Given the current study results, we recommend that seqB, ID, and plug-n-play methods be included in any suite. plug-n-play does not require knowledge of the serial distribution and provides close to true estimates under different model structures quickly. seqB and ID should be implemented using a range of known serial intervals, to provide sensitivity analysis and confidence in $R_0$ estimation. We do however note that plug-n-play may be difficult to implement for some, since the R package is quite technical [45].

Daily case reporting data has been available for the most recent COVID-19 pandemic. Daily data was not provided during the 2009 H1N1 pandemic, however. Furthermore, there may be issues with daily reporting (such as periodicity, reporting delay) whereby public health may choose to use weekly reporting data over daily data as the weekly data would be more reliable. We have thus only considered weekly case reporting data in this study as it is expected that weekly case reporting data can be expected in many future epidemics and pandemics. It is important to note that First Few Hundred (FF100) studies, whereby the first few hundred cases of a new virus are followed in detail at the beginning of an infectious disease outbreak, have been implemented during the 2009 H1N1 and COVID-19 pandemics [53–60]. In these cases the serial distribution, and the need to consider exposed and/or asymptomatic periods of infection can be quickly determined, enabling realization of earlier and more certain estimates of $R_0$ early on. Given that First Few Hundred protocols are not implemented in much of the globe, weekly case report data however may still be considered the norm for future pandemics.

In our current study we have assumed perfect data with no unobserved infections, no reporting delay, and no data collection bias. These issues are intuitively expected to affect $R_0$ estimates. We venture to continue our study of $R_0$ estimation considering these aspects in our epidemiological data sets.

In summary, our work has various strengths, and some limitations. A unique strength of our work is the study of model misspecification. We are unaware of previous work in this direction. We did not consider all possible estimators of $R_0$, but focused on those most commonly used in the field of Infectious Disease Modelling. We selected a variety of influenza and COVID-19 scenarios for our simulations, which provide considerable information on the behaviour of these estimators. We did not investigate other infectious diseases, such as Ebola, which could potentially have quite different parameters. Our overall recommendations are however, general, and are therefore widely applicable. Lastly, we considered only the scenario of perfect data. Alternative settings are beyond the scope of this work, however, this, along with other infectious diseases and potentially more estimators will be considered in future.

## Supporting information

**S1 File. A supplementary file contains additional simulations results (both tables and box-plots) as well as some further technical details.**
(PDF)

**S1 Appendix.**
(PDF)

## Author Contributions

**Conceptualization:** Sawitree Boonpatcharanon, Jane M. Heffernan, Hanna Jankowski.

**Data curation:** Jane M. Heffernan.

**Formal analysis:** Sawitree Boonpatcharanon, Jane M. Heffernan, Hanna Jankowski.

**Funding acquisition:** Jane M. Heffernan, Hanna Jankowski.

**Investigation:** Sawitree Boonpatcharanon, Jane M. Heffernan, Hanna Jankowski.

**Methodology:** Sawitree Boonpatcharanon, Jane M. Heffernan, Hanna Jankowski.

**Project administration:** Jane M. Heffernan, Hanna Jankowski.

**Resources:** Sawitree Boonpatcharanon, Jane M. Heffernan, Hanna Jankowski.

**Software:** Sawitree Boonpatcharanon, Jane M. Heffernan, Hanna Jankowski.

**Supervision:** Hanna Jankowski.

**Validation:** Sawitree Boonpatcharanon, Jane M. Heffernan, Hanna Jankowski.

**Visualization:** Sawitree Boonpatcharanon, Jane M. Heffernan, Hanna Jankowski.

**Writing – original draft:** Sawitree Boonpatcharanon, Jane M. Heffernan, Hanna Jankowski.

**Writing – review & editing:** Sawitree Boonpatcharanon, Jane M. Heffernan, Hanna Jankowski.

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
