## [Decision Letter · Decision Letter 0]

8 Oct 2021

PONE-D-21-22343Estimating the basic reproduction number at the beginning of an outbreak under incomplete dataPLOS ONE

Dear Dr. Heffernan,

Thank you for submitting your manuscript to PLOS ONE. After careful consideration, we feel that it has merit but does not fully meet PLOS ONE’s publication criteria as it currently stands. Therefore, we invite you to submit a revised version of the manuscript that addresses the points raised during the review process.

We look forward to receiving your revised manuscript.

Kind regards,

Inés P. Mariño, Ph.D.

Academic Editor

PLOS ONE

Journal Requirements:

Reviewers' comments:

Reviewer's Responses to Questions

**Comments to the Author**

1. Is the manuscript technically sound, and do the data support the conclusions?

Reviewer #1: Partly

Reviewer #2: Partly

2. Has the statistical analysis been performed appropriately and rigorously? 

Reviewer #1: I Don't Know

Reviewer #2: Yes

3. Have the authors made all data underlying the findings in their manuscript fully available?

Reviewer #1: No

Reviewer #2: Yes

4. Is the manuscript presented in an intelligible fashion and written in standard English?

Reviewer #1: Yes

Reviewer #2: No

5. Review Comments to the Author

Reviewer #1: In “Estimating the basic reproduction number at the beginning of an outbreak under incomplete data” by Boonpatcharanon and colleagues, different methods to estimate R0, the basic reproduction number, are compared considering the first 100 days of an epidemic. The authors apply frequentist and Bayesian approaches to estimate R0 under three different infection models: SIR, SEIR and SEAIR. These models differ in the allowed transitions between states of individuals (susceptible, exposed, (asymptomatic/symptomatic) infected, recovered). The authors conclude with a recommendation but also highlight that it always depend on the data which approach to choose; they recommend sensitivity analyses.

The aim of the study is described and motivated. The manuscript is well written. However, there are some issues the authors should consider to facilitate the readability of the manuscript.

Major issues:

1) “Incomplete data” sounds like missing data related to counts of, e.g., infected individuals. Should “incomplete” also comprise incomplete knowledge/information on transmission and course of infection? Please clarify (in the manuscript and probably in the title). Additionally, please add information on the required (observed) data underlying the R0 estimation/calculation.

2) Please provide real data applications to support the assumptions in the simulation study and to illustrate the investigated methods on real data. For influenza, weekly case reports are published for several seasons, for example by the ECDC (European Centre for Disease Prevention and Control), the Government of Canada or the CDC (Centers for Disease Control and Prevention).

3) Please provide the (documented) source code for the investigation to redo the analysis (including data simulation and figure/table preparation).

4) Introduction:

a. Could the authors elaborate more on data misspecification? Maybe through an own paragraph including examples of misspecifications and their possible influence on the R0 estimates? This issue is related to the reliability of an R0 estimation in the epidemic situation itself. The benefit/value of the R0 estimation depends heavily on the population under investigation, i.e. whether this population is a random sample of the total population or a for the total population not representative subpopulation (i.e. comprising, e.g., more or fewer infected individuals or different transmission probabilities than in the total population). Issues to be considered are for example the test strategy (which individuals are tested or must provide a test result; related to the number of unreported cases) and the test quality (reliable test results).

b. As the study is about the early stage of an epidemic (first 15 weeks), could the authors additionally include this time frame into the considerations about misspecification? In case of a “new” disease, the knowledge on which, e.g., R0 estimation is based is limited in the early days. Could the authors please highlight the important issues unique to the beginning of an epidemic/pandemic – compared to the subsequent time? Besides in the beginning of an epidemic, is it also possible to consider a time point within an epidemic with a very low number of infected individuals, e.g. between two waves or two seasons (in case of seasonality as for influenza)? Please clarify “early stage”. Please add a motivation for considering only the first 15 weeks.

c. Please add a motivation for the decision to consider SIR, SEIR and SEAIR only.

5) Materials and Methods:

a. Please provide the underlying assumptions related to the data for the investigation (i.e. no unobserved infections, no reporting delay, …).

b. Please include a section about the simulation study. The approach description should not be part of the result section and the parameter choice should not be part of the method description. Please aggregate.

c. In some parts, methods are provided in the results section and vice versa. Please check and separate.

d. Lines 64-77: Please provide a supporting figure for illustration, if possible. Furthermore, please consider the inclusion of Table 1 in this figure and, if possible, remove Table 1.

e. Line 97: Please introduce the methods briefly (including the reference to the respective subsection) and provide the abbreviations used throughout the manuscript. Then, refer to Table 3. Otherwise, the subsequent sections cannot be followed easily.

f. Line 103: Please consider to describe “serial distribution” earlier in the manuscript because it was already used earlier. Suggestion: Provide a section with definitions needed for the models (SIR, SEIR, SEAIR). Furthermore, please consider a summarisation of all parameters that are set to some selected values in the investigation. A table (or subheadings after re-ordering) might help.

g. Part 0.2.1

i. Please check notations and definitions. For example:

1. Line 133: “or” instead of “, or,“.

2. Please unify kappa and k.

3. Line 135: “both” does not fit to “the method”, which is one method. Please check.

4. Lines 137/138: Please add the origin for “number of days or weeks”.

5. Line 139: Please clarify min(kappa, t). What is t?

6. Line 139: Please clarify the relation between I(t – t_j) and I(t), if there is one, otherwise please define I(time difference / interval).

7. Line 159: Please clarify “built-in alternative optimisation”. Where is it “built-in”?

ii. Please provide p(t_j) for all models.

iii. Lines 149/150: Please explain the limitation.

iv. Lines 150: Please provide the section reference for the simulations.

v. Instability issues (lines 156-165):

1. Might the instability be an indicator for non-adequateness of the applied method?

2. Please consider to include the observed instability issues in the result section to clearly separate methods and results (introduction of new subsection headings might help). Is it possible to quantify these issues?

3. Was the implementation of the grid search approach in comparison to the original implementation validated? If so, how?

h. Part 0.2.2:

i. As long intervals without new infections are problematic for this approach, this approach might be better suited for situations after the start of a new “wave” with rapidly increasing numbers of newly detected infections. Did the authors investigated scenarios, in which the numbers only increased slowly, or were the scenarios adapted to this method? In the latter case, a comparison in a non-adequate scenario would be of interest to guide future method applications. Especially in the beginning of a pandemic, such situations might occur.

ii. Lines 214/215: Please state the adaptations in more detail. Was the implementation in comparison to the original implementation validated? If so, how?

i. Part 0.2.3:

i. Lines 233/234: Please clarify “beginning of an outbreak”. The authors state that the number of infectious individuals rapidly decreases in the beginning, but in the beginning of a new disease few individuals are infected/infectious and the number of infected/infectious people increase. Otherwise, I would expect that R0 is overestimated as the estimate does not decrease fast enough. Please clarify.

ii. Lines 244/245: Please provide a reference to the specifications of the misspecification.

j. Part 0.2.4:

i. Please provide (throughout the manuscript) names of R packages besides the reference.

ii. Line 275: Please explain “particle”.

iii. Equation after line 279: R0 is probably not a single value as delta_t is probably a sequence. Please check and adapt, if necessary.

iv. Line 280: Please clarify where “regardless of the epidemiological model” relates to (and what is model-dependent).

v. Line 282: Please check the reference to the appendix. Appendix 1.3 is “Least square estimation for the IDEA method”. Please provide more comments in the source code (Appendix 1.4) and please check line breaks to facilitate reading.

k. Part 0.2.5:

i. Line 292: Please provide the respective simplifications in the subsequent derivations.

ii. Lines 294/295: Please describe m more clearly. Please explain additionally (besides the equation) m_j in words. Definition of m0 should be provided with the definition of m_j.

iii. Line 295: Please clarify “epidemic” and “much more information”.

iv. Lines 296/297: Please check the conditions for i.

v. Lines 299/300: What is the impact, if an individual needs more than one week to recover? What is the motivation for one week? Please add.

vi. Line 334: “obtained” instead of “obtain”

6) Results:

a. Lines 351-353: Please additionally consider the case that the population studied is not a random sample of the target population. Alternatively, please clearly state (when defining the study design) the assumption that the populations studied is a random sample and discuss this assumption as limitation.

b. Lines 377 to 380: Does the results change if the other methods are also only applied to the subset of samples? Please comment.

c. Lines 380/381: Please define bias and variability. Did the authors also consider a joint measure of bias and variability?

d. Line 382: A figure cannot study. Please rephrase throughout the manuscript.

e. Please consider to add further subsections to provide more guidance to the reader.

f. Line 405: Computation time is provided but the related section follows later-on. Please reorder.

g. Part 1.1: Could the authors please provide computational aspects for all models?

7) Discussion:

a. Please provide a paragraph about strength and limitations.

b. Please compare the results (at least in parts) with other studies.

8) Abbreviations, parameter, model names, methods names and other short forms:

a. Please introduce all in the main part of the manuscript. E.g. ODE, MCMC, IID, S0, I0, S, I, S(t), SD, … are missing.

b. Please check the usage for consistency, e.g. S versus S(t).

c. Please state which parameter are 0 at t=0.

9) Figures:

a. Please provide axis titles at the respective axis and not in the description.

b. In case the legend only comprises one symbol/colour differing between figure panels, please consider providing this information as panel title above the respective plot panel. This also introduces shorter description.

c. Please introduce all abbreviations, parameter and model names in the figure description.

d. In case of boxplots, please provide complete boxplots. In case of a needed zoomed-in boxplot, the complete one should be provided in the supplement.

e. Please provide information in the description of the boxplots so that the reader is able to identify scenarios with misspecifications.

10) Tables:

a. Please introduce all abbreviations, parameter, model names and method names in the table description.

b. Please provide a description that allows to understand the table without the part in the main manuscript where the table is cited for the first time.

Minor issues:

1) Section numbering in the main part: Please remove the leading “0.”. Please check the complete numbering and doubling of section headings, e.g. “Results” and “1. Results” and supporting information starts with 1.2.

2) Please consider to avoid “flu” and to use “influenza” throughout the manuscript.

3) Materials and methods:

a. Line 58: Please clarify “approximately”.

b. Line 77: It should probably be I(0) = 1 (first round bracket is misplaced).

c. Line 85: Please provide information on the meaning of “inflection” in lay terms (i.e. related to the course of infection/pandemic).

d. Line 126: Please provide some additional information on the computer.

e. Part 0.2.2:

i. Equation after line 192: To stick to the notation throughout the manuscript, please consider replacing s by t, i.e. S(t) and dt.

ii. Line 194: Please consider to replace | by “given”, i.e. “conditional distribution of I(t_j+1) given I(t_j) and R_0”. This would facilitate reading.

iii. Line 196: Please introduce N0.

f. Part 0.2.3:

i. Please introduce s and d.

ii. Line 230: Please delete “obvious”.

iii. Equation (4): Please consider to use additional brackets so that it is clear to which the sum sign belongs.

iv. Lines 243/244: “However, …” instead of “…, however.”.

4) Figure 1:

a. Lines 84/85: Please consider to remove parts of figure descriptions from the main text that should be part of the description accompanying the respective figure itself, i.e. below the figure panel(s).

b. Please introduce the meaning of “inflection”.

5) Table 2: Please clarify the meaning of Y_i (exponentially distributed with a mean of 1). Later-on, it is a mean of 1/gamma (provided as an example). Or other natural numbers. Please consider a consistent notation.

6) Supporting information:

a. Part 1.2:

i. Please provide references for the models and their chosen parametrisation.

ii. Please introduce all parameter in more detail, even if they are introduced in the main text. Providing all definitions facilitates reading. The authors could consider to introduce a separate section within 1.2 for definitions. An alternative might be to provide the definitions in the main text, e.g. in a table.

b. Part 1.3:

i. Please provide the partial derivatives and few more steps of the solving process.

Reviewer #2: This manuscript describes an interesting simulation study comparing 6 different methods of estimating the R0 coefficient (WP, secB, ID, IDEA, plug-n-play and fullBayes). The data are simulated via three different compartmental models, SIR, SEIR and SEAIR. Methods are intended to be tested both under the well-specified model and parameters and under the miss-specified ones. The quality of this work is the large range of methods tested, from the more classical and simplified models to the fully Bayesian ones. However, while the idea of comparing the performance of the methods is good and promising and the spectrum of methods compared is broad, the study and manuscript suffer from several weaknesses.

The biggest problem is a misunderstanding of two random duration variables involved in the epidemiological analysis of a pandemic: the infectious period and the serial interval. The first is the random length of time a subject remains infectious, the second is the random time between when the infector develops symptoms and when the infected develops symptoms in a chain of transmission (see for example: Zhou X-H, You C, et al, 2020, the Lancet). These two intervals are in general quite different in mean; for instance for COVID-19 infection the mean infectious period is around 8-10 days (He X, Lau EHY, et al 2020, Nature; Zhou X-H, You C, et al, 2020, the Lancet) while the mean serial interval is around 4-5 days (Nishiura et al 2020, IJID; Du et al, 2020, CDC; Zhou X-H, You C, et al, 2020, the Lancet). The mix-up between these two intervals (and distributions) is evident on page 4 when it says: “The serial distribution is the distribution of the random amount of time that an individual is infected..”.

This inaccuracy has consequences for the simulation study. In fact, data generated according SIR model of parameters beta and gamma have by construction mean infectious period of 1/gamma (fixed at 5 days for simulations). The problem arises when methods adopted for R0 estimation depend on the serial interval distribution, instead of the infectious period distribution, which is the case of the WP (White and Pagano 2007), ID and IDEA (Fisman 2013). In these cases models will not be well specified even when authors present them as being so. This can explain why in Fig 5, for example, WP, ID and IDEA methods (lines 1,3 and 4) seem to perform better when the gamma parameter is incorrect (right panel) than when it is correct (left panel). And comparing Fig 5 and 6 for the same methods, performance is improved when the model is miss-specified (R0 estimated assuming SIR with SEIR data). The authors need to address this point first.

A second point is inherent in the design of the simulation and the presentation of the results. The data are indeed simulated under a single choice of parameters, which may not be sufficient to draw general conclusions. Here, the parameters are chosen with respect to a given infection (influenza). It seems to me that adding other parameter choices would add value to the study. In addition, attention should again be paid to the fact that the gamma parameter do refer to the distribution of the infection period and not to the distribution of the serial interval.

The results are presented by boxplots, which is a good idea. However, on the one hand, some graphs are repeated several times (e.g. the WP case (SD = exp mean 5/7) with the SIR data is repeated 3 times in fig 2, 3 and 5), and I believe that a way could be found to avoid this. On the other hand, the results should also be presented numerically in tables, with for each setting the specification of the bias and variability of the simulated results at the inflection point, or with a summary of both (mean square error).

An application to real data would also be interesting, in order to see how different R0 estimations the considered methods can produce on observed incidence data. I would personally be interested in seeing these results for COVID-19 outbreak.

Finally, a thorough review of the English language is necessary.

Specific points:

Page 2, line 26. ….”serial interval, infectious period… “. Please define all quantities when they are introduced

Page 3, line 74. Here gamma is set to 1/3, while in the Result section it is set to 1/5 (or 7/5 with weekly data).

Page 3, line 100. “ODE epidemiological model “. Please define

Page 12, line 391: “Note that here the mean of the serial distribution was incorrect by only two days….”. Here authors don’t comment the fact that performance is better with the wrong serial distribution (see my comment above). In addition the amount of miss-specification (2 days) is chosen by the authors and they can modify it if it seems not enough to show some effect. I recommend testing a range of parameter choices.

Page 15, line 490-91. “Asymptomatic infected (infected, no symptoms, not infection)”. Replace with : (infected, no symptoms, infection)

6. PLOS authors have the option to publish the peer review history of their article (what does this mean?). If published, this will include your full peer review and any attached files.

Reviewer #1: **Yes: **Miriam Kesselmeier

Reviewer #2: No

---

## [Author Response · Author response to Decision Letter 0]

25 Jan 2022

We thank the reviewers for their comments. We have added new examples to our study. We have also revised the manuscript for enhanced clarity and understanding. We have provided a detailed response to reviewers as an attachment.

---

## [Decision Letter · Decision Letter 1]

4 Mar 2022

PONE-D-21-22343R1Estimating the basic reproduction number at the beginning of an outbreakPLOS ONE

Dear Dr. Heffernan,

Thank you for submitting your manuscript to PLOS ONE. After careful consideration, we consider that the manuscript is much improved but still does not fully meet PLOS ONE’s publication criteria. Therefore, we invite you to submit a revised version of the manuscript that addresses the points raised by one of the reviewers.

We look forward to receiving your revised manuscript.

Kind regards,

Inés P. Mariño, Ph.D.

Academic Editor

PLOS ONE

Reviewers' comments:

Reviewer's Responses to Questions

**Comments to the Author**

1. If the authors have adequately addressed your comments raised in a previous round of review and you feel that this manuscript is now acceptable for publication, you may indicate that here to bypass the “Comments to the Author” section, enter your conflict of interest statement in the “Confidential to Editor” section, and submit your "Accept" recommendation.

Reviewer #1: (No Response)

Reviewer #2: All comments have been addressed

2. Is the manuscript technically sound, and do the data support the conclusions?

Reviewer #1: Yes

Reviewer #2: Yes

3. Has the statistical analysis been performed appropriately and rigorously? 

Reviewer #1: Yes

Reviewer #2: Yes

4. Have the authors made all data underlying the findings in their manuscript fully available?

Reviewer #1: No

Reviewer #2: Yes

5. Is the manuscript presented in an intelligible fashion and written in standard English?

Reviewer #1: Yes

Reviewer #2: Yes

6. Review Comments to the Author

Reviewer #1: The authors have greatly improved their manuscript. The additional explanations facilitate reading very much.

However, some issues remain to be solved or should at least be considered:

1) Source code: For reasons of reproducibility, please make the complete source code including the C++ code available.

2) Abbreviations:

a. Please introduce all abbreviations (e.g., SI in the text and IF in table 3).

b. Please consider avoiding some abbreviations (e.g., SI or SD) for a better readability of the text. SI and SD are only used in some places and, hence, might be avoided. This also applies to IF in line 498.

3) Citations:

a. Please introduce space between authors and “(Year)”, e.g. “Anderson and May (1992)” or “Allen et al. (2008)”.

b. Please remove “[“ and “]” if the citation is a real part of the text and not only a reference to the reference section. Examples: Lines 23/24 or line 54. The brackets should remain, e.g., in lines 80/81 or line 134.

4) Reference to supplement: Could the authors please provide a more detailed reference to the supplement in the main text, e.g. “see supplemental section 1.1” or “Fig. 7”?

5) Supplement, lines 608-611: Please provide more calculation steps, such that the reader can follow the derivations more easily. This is not for understanding the method but for enabling an easy tracking of the derivations.

6) Please stick to one notation – either influenza 1 and influenza 2 or influenza one and influenza two (main part and supplement).

7) Abstract: Could the authors please add some results and a conclusion to the abstract?

8) Introduction: The authors clarified „early stage“. However, I would like to come back to a point I made on the previous manuscript version. The authors state, that they only consider a single wave. Does this imply that (i) the complete pandemic only runs for one wave, (ii) each wave is considered as a new pandemic or (iii) that only the first wave is considered in this manuscript? This question arises, as the authors seem to model influenza seasons as different pandemics (or as different waves). At least for me, a detail is missing. Is there a difference between “wave” and “season” in the application of methods? Please clarify.

9) Methods:

a. Language:

i. Line 105: “equal” instead of “equation“

ii. Line 143: “for our models” or “for the models” instead of “for the our models”

iii. Lines 156/157: Please rephrase “The main difficulty in estimation is that complete data is unavailable for the full epidemiological model is unavailable.” Delete “is unavailable” at the end of the sentence or replace “for” by “if”?

iv. Line 190: Please delete “is a”.

v. Line 196: “)” missing after t_(j+1).

vi. Line 239: I would like to emphasize, that the word “given” instead of “|” would facilitate reading as it is provided in the text and not in a formula. Furthermore, there might be readers that are not so familiar with mathematical notations, but want to read the method section.

vii. Line 353: “…” between “b+1” and “b+B” probably missing.

viii. Line 357: “obtain” instead of “obtained”

ix. Line 363: “obtained” instead of “obtain”

x. Lines 370-373: Is “…”necessary?

b. Format:

i. Lines 106-108: Please consider providing each equation in a separate line instead of in line with the remaining text.

ii. Lines 153, 296, 348: Section number is missing. Please check throughout the manuscript.

c. Definitions / settings:

i. Line 105: Please consider “for all time points t >= 0” instead of “for all t >= 0”.

ii. Line 192: Please consider “t_0 = 0 (beginning of the pandemic), t_1, …” (or something similar) instead of “t_0 = 0, t_1, …” to provide the time origin.

iii. Equation between lines 237 and 238 as well as line 238: Above, t >= 0 is time in the process. Maybe I am missing something, but I would expect an subscript at t as it is related to I(t_(j+1))

iv. Equation between lines 276 and 277: Probably, s_j is similarly defined as t_j. Please state this briefly.

v. Line 335: Please introduce theta (again).

vi. Line 351: Please introduce B.

d. Please consider providing the link to the github somewhere in the main manuscript.

10) Results:

a. A figure or lines cannot plot (e.g. lines 441-443). Information is, e.g., provided or indicated in a figure. Please rephrase and check throughout the manuscript.

b. Line 449: The comma after the 9 is probably misplaced. Please check.

c. Line 450, suggestion: Replace “5, 6” by “5 and 6” to be consistent with the previously provided numbers.

d. Lines 502-506: Probably “if” instead of “when”.

11) Discussion:

a. Please provide a paragraph about strength and limitations (and approaches to mitigate them) of the investigation (not of the studied estimators) presented in this manuscript – maybe just by reordering of the paragraphs or by highlighting these issues.

b. Line 549: In the discussion, results provided in the result section should be summarised and a reference to the supplementary material should not be necessary.

12) Figures (main part and supplement):

a. In general:

i. Please assure readability of all parts of the plot, including axes and legend.

ii. Please provide axes names (including unit) on the respective axes and not only in the figure description below the plot.

iii. Please avoid overlapping plot symbols in the legend.

iv. Please introduce for each figure all used abbreviations, e.g. SI.

b. Additionally in Figure 1: The legends could be omitted, if the model is provided in the top of each column – similarly to the disease in front of the rows. This would facilitate reading.

c. Additionally in Figure 6:

i. Suggestion: Providing the column information (Canada, provinces) above the columns might facilitate reading, especially in the presence of the axes names (see 12.a.ii).

ii. Lines 517-519 should be part of the figure 6 description itself.

d. Additionally in Figure 8:

i. The legend could be omitted, if the data is provided in front of each row.

ii. Please provide the prior distribution above each column of the plot.

13) Tables (main part and supplement):

a. In general: Please introduce all abbreviations, e.g. SI.

b. Table 1 (c): Please refer to (b) for tuple definition.

c. Table 2: No reference for ID method available?

d. Supplement: Please correct “… denotes a standard deviation great than…” to “denotes a standard deviation greater than”.

Reviewer #2: Thank you for having addressed all of my comments regarding the description of the method and presentation of the results. The manuscript is now easier to understand, thanks also to a revision of the English language.

7. PLOS authors have the option to publish the peer review history of their article (what does this mean?). If published, this will include your full peer review and any attached files.

Reviewer #1: **Yes: **Miriam Kesselmeier

Reviewer #2: No

---

## [Author Response · Author response to Decision Letter 1]

18 Apr 2022

We have supplied a rebuttal letter that responds to each point raised by the academic editor and reviewer(s). This is uploaded as a separate file labeled 'Response to Reviewers'.

We have supplied a marked-up copy of our manuscript that highlights changes made to the original version, labeled 'Revised Manuscript with Track Changes'.

We have uploaded an unmarked version of our revised paper without tracked changes, labeled as 'Manuscript'.

---

## [Decision Letter · Decision Letter 2]

4 May 2022

PONE-D-21-22343R2Estimating the basic reproduction number at the beginning of an outbreakPLOS ONE

Dear Dr. Heffernan,

Thank you for submitting your manuscript to PLOS ONE. After careful consideration, we feel that it has merit but does not fully meet PLOS ONE’s publication criteria as it currently stands. Therefore, we invite you to submit a revised version of the manuscript that addresses the points raised during the review process.

We look forward to receiving your revised manuscript.

Kind regards,

Inés P. Mariño, Ph.D.

Academic Editor

PLOS ONE

Journal Requirements:

Reviewers' comments:

Reviewer's Responses to Questions

**Comments to the Author**

1. If the authors have adequately addressed your comments raised in a previous round of review and you feel that this manuscript is now acceptable for publication, you may indicate that here to bypass the “Comments to the Author” section, enter your conflict of interest statement in the “Confidential to Editor” section, and submit your "Accept" recommendation.

Reviewer #1: (No Response)

2. Is the manuscript technically sound, and do the data support the conclusions?

Reviewer #1: Yes

3. Has the statistical analysis been performed appropriately and rigorously? 

Reviewer #1: Yes

4. Have the authors made all data underlying the findings in their manuscript fully available?

Reviewer #1: Yes

5. Is the manuscript presented in an intelligible fashion and written in standard English?

Reviewer #1: Yes

6. Review Comments to the Author

Reviewer #1: The authors have addressed almost all of my comments.

An issue that was not addressed, yet, can be found in lines 410-425. There are still "influenza one" and "influenza two", although the authors stated that they have changed all to "influenza 1" and "influenza 2", respectively. Please adapt.

Coming back to strength and limitations in the discussion, my wording was not clear. I am sorry for the inconvenience. My suggestion was to additionally provide a paragraph on strengths and limitations (and approaches to mitigate them) of the conducted study with respect to, e.g., selected scenarios and the selection of the investigated estimators. Additionally to the strengths and limitations of the investigated estimators. Maybe a simple reordering might be a solution for highlighting.

7. PLOS authors have the option to publish the peer review history of their article (what does this mean?). If published, this will include your full peer review and any attached files.

Reviewer #1: **Yes: **Miriam Kesselmeier

---

## [Author Response · Author response to Decision Letter 2]

12 May 2022

Dear Editor and Reviewers,

We thank the Editor and both reviewers for their consideration and careful reading of our manuscript. Reviewer #2 is now completed satisfi\fed with the manuscript and is not requesting a single change. Reviewer #1 has requested two minor changes and we have implemented both. 

Comment 1: The authors have addressed almost all of my comments.

(a) An issue that was not addressed, yet, can be found in lines 410-425. There are still "influenza one" and "influenza two", although the authors stated that they have changed all to "influenza 1" and "infuenza 2",

respectively. Please adapt.

We missed these entries in the previous revision. This has been done.

(b) Coming back to strength and limitations in the discussion, my wording was not clear. I am sorry for the inconvenience. My suggestion was to additionally provide a paragraph on strengths and limitations (and approaches to mitigate them) of the conducted study with respect to, e.g., selected scenarios and the selection of the investigated estimators. Additionally to the strengths and limitations of the investigated

estimators. Maybe a simple reordering might be a solution for highlighting.

We have summarized some strengths and limitations in a final paragraph to the Conclusion.

---

## [Editor Report · Decision Letter 3]

19 May 2022

Estimating the basic reproduction number at the beginning of an outbreak

PONE-D-21-22343R3

Dear Dr. Heffernan,

We’re pleased to inform you that your manuscript has been judged scientifically suitable for publication and will be formally accepted for publication once it meets all outstanding technical requirements.

Kind regards,

Inés P. Mariño, Ph.D.

Academic Editor

PLOS ONE
---

## [Editor Report · Acceptance letter]

7 Jun 2022

PONE-D-21-22343R3 

Estimating the basic reproduction number at the beginning of an outbreak 

Dear Dr. Heffernan:

I'm pleased to inform you that your manuscript has been deemed suitable for publication in PLOS ONE. Congratulations! Your manuscript is now with our production department. 

Kind regards, 

on behalf of

Dr. Inés P. Mariño 

Academic Editor

PLOS ONE